# Network Depth and Inductive Bias
# in Convolutional Neural Networks

## Abstract

Understanding the remarkable generalization abilities of Deep Learning systems remains one of the significant scientific challenges of our time. It is widely accepted that the success of DNNs stems, at least partially, from having many hidden layers. However, the benefits of such depth are not universal. We introduce a simple experimental paradigm that demonstrates the contrasts between CNNs and MLPs in this respect. This paradigm demonstrates the ability of contemporary architectures to leverage deep, multi-layered structures to systematically improve model generalization ability. However, this conflicts with statistical learning theory (SLT) and its key concept of the bias-variance tradeoff. Therefore, we present an alternative framework to understand the relationship between network architecture and generalization by viewing classifiers as maps between different metric spaces. Through comparative analysis, we uncover how deeper networks develop a bias towards smoother input representations; and that the inductive bias responsible for superior generalization in deep CNNs is distinct from the standard "minimal complexity" (Occam's razor) that is the focus of SLT.

## 1 Introduction

Generalization is the ability of a trained model to perform well on unseen data. In supervised machine learning, labelled data is used to train an algorithm to classify data or predict a certain outcome. As training data is input into the model, it compares the output to the target through some loss function. For neural networks, the error of the loss function is propagated through the network to adjust the model parameters to minimize the error. This process is tailored to find the best solution for the data at hand (optimization) and unseen data (generalization). Since performance on unseen data is the central goal of pattern recognition, understanding its origin and characteristics is the main concern of theoretical research in Machine Learning.

Large Language Models (LLMs) are currently the most visible application of Artificial Intelligence (AI), and the transformer architecture (Vaswani, 2017) is at the core of all leading LLMs. Transformers were developed to address the sequence-to-sequence mapping task that arises in applications such as machine translation and proved to be extremely powerful in this regard. Researchers quickly found ways to cast a wide range of applications as sequence-to-sequence tasks: not only language modelling and speech recognition but also computer vision and even robotics have seen impressive transformer-based developments.

Consequently, much of the theoretical focus in Artificial Neural Networks (ANNs) has switched to transformer architectures, even while fundamental issues with earlier architectures such as Convolutional Neural Networks (CNNs) are not adequately understood. Most importantly, the CNN literature contains interesting hints (Zhang et al., 2021) that the core principles in Statistical Learning Theory are not relevant to the excellent performance of state-of-the-art CNNs: over-parametrized networks may perform optimally, in conflict with the classical bias-variance trade-off (Geman et al., 1992) which is central to Statistical Learning Theory. This conflict is related to the fact that deep CNNs (those with many hidden layers) significantly outperform shallower networks – something that is not true for earlier networks such as multilayer perceptrons (MLPs).

Given that CNNs are still practically important (especially in applications such as computer vision), and that insights on CNN generalization may also be useful for understanding transformers and future architectures, we investigate this apparent conflict from a novel angle. In particular, we present additional evidence of the contrast between CNNs and MLPs concerning the number of computational layers and then study the root causes of this contrast.

The main contributions of this work are threefold. We describe a simple experimental test bed that allows us to study the elementary drivers of generalization in CNNs. We also introduce a precise expression for the end-to-end piecewise linear transfer function of CNNs with Rectified Linear Unit (ReLU) (He et al., 2015) transfer functions, which allows one to apply all the machinery of linear algebra in each linear patch of a CNN. Finally, we use these tools to show that the inductive bias responsible for superior generalization in deep CNNs is quite distinct from the standard "minimal complexity" (Occam's razor) that is the focus of Statistical Learning Theory.

## 2 Related Work

In this section, we briefly review the classical Statistical Learning Theory (SLT) (Vapnik, 1999) perspective on the factors that influence the generalization ability of deep neural networks and then summarize an example of more modern work that transcends the limitations of SLT. Section 2.1 introduces the concept of bias-variance trade-off and its relationship with model complexity. Section 2.2 discusses how the sharpness of the loss function can influence the generalization depending on its smoothness. Section 2.3 briefly highlights the various factors that determine classifier complexity in practice. Section 2.4 contextualises the concept of interpolation and its applicability.

### 2.1 Bias-variance trade-off

The basic concept of bias-variance tradeoff (Geman et al., 1992) refers to the compromise between a model's ability to correctly capture the intrinsic data patterns and its responsivity to changes in the training data. The first factor, bias, is the average gap between the model's prediction and the target value. In contrast, the second, variance, can be seen as the average variability of a model's prediction for equivalent conditions (e.g., different data set samplings or training conditions). Complex machine learning models extract complex and convoluted patterns in the training data but can, in turn, model the noise, which results in high variance. Conversely, an oversimplified model representation is biased because it fails to capture the complexity of the data.

The bias-variance trade-off suggests that as the complexity of a model increases, its variance also increases but its bias decreases. Conversely, as the model complexity decreases, the variance decreases but the bias increases. This balance of complexities leads to a U-shaped test error curve depicted in Figure 1.

According to SLT, the ultimate aim is to strike a balance or compromise between the two extremes, shown between the dotted lines (optimal zone). This ensures that a model performs well on unseen data and generalizes well. As shown in Figure 1, a model that operates to the left of the optimal zone has high bias, is oversimplified, does not learn adequately from the training data and is said to underfit. In contrast, a model that operates to the right of the optimal zone has high variance and performs well on training data but poorly on unseen data and is said to overfit. Classical wisdom highlights that the trade-off is intrinsically associated with model complexity. In this frame of reference, finding a model with sufficient capacity to achieve the optimal bias-variance trade-off is necessary.

Fortmann-Roe (2012) provided experimental evidence highlighting the bias-variance trade-off for non-parametric methods such as k-Nearest Neighbors and kernel regression in common classification and regression settings. For neural networks, this suggests that increased network width leads to decreased bias, while the variance increases. However, modern neural network architectures have frequently demonstrated the convenience of having many more parameters than classic wisdom dictates (Krizhevsky et al., 2012; Simonyan & Zisserman, 2014). This contradiction between modern practices and conventional wisdom suggests that some aspects of the classical theory may not be appropriate for understanding modern networks.

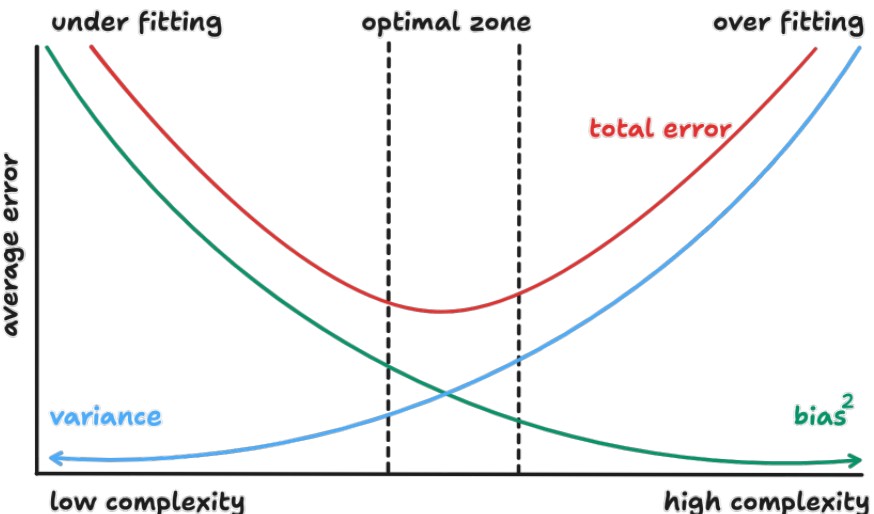

Figure 1: Bias-variance tradeoff plot (Fortmann-Roe, 2012) shows low complexity model with high bias turns to underfit the data while a high-complexity model with high variance tends to overfit. However, a zone exists between the dotted lines where the model will likely generalize well.

Thus, recent works show that overparameterized networks do not exhibit the classic U-shaped error curve. This is demonstrated by the decrease in the test error for wider networks (Neyshabur et al., 2014; Novak et al., 2018; Belkin et al., 2019). The absence of the U-shaped curve may imply the presence of some implicit regularization that regulates network capacity (Neyshabur et al., 2017), as we discuss in Section 2.3). However, a simple test error analysis does not reveal the reasons behind the absence of the bias-variance trade-off. Research by Neal et al. (2018) showed that the variance due to optimization increased significantly in low-complexity spaces but decreased monotonically with width in high-complexity spaces.

## 2.2 Sharpness in the weight landscape

Empirical evidence has shown that Stochastic Gradient models that converge to flatter minima tend to generalize better than those that converge to sharp minima. Among others, Hochreiter & Schmidhuber (1997) define a flat minimum as a wide region in weight space where every weight vector from that region results in a comparable error. To highlight this point, regions with flat minima only require low-precision weights compared to high precision for sharp minima. Flat minima are less complex, require fewer bits to describe and based on Minimum Description (Message) Length theory, generalize better (Wallace & Boulton, 1968).

Keskar et al. (2016) empirically observed that large-batch methods are drawn to regions of sharp minima, which resulted in the degradation of a model's ability to generalize. Conversely, small-batch methods are attracted to wider and flatter minima, as shown in Figure 2. The figure also shows that flat minima are insensitive to variations and provide robustness to random perturbations, an essential characteristic of a deep network. The low generalization performance of large batch methods can be attributed to the high number of positive eigenvalues in the Hessian matrix $\nabla^2 L(w^*)$. This contrasts methods with a few positive eigenvalues that tend to generalize well. While this work primarily focused on the maximum loss within the restricted region of the minima, Chaudhari et al. (2019) proposed the Entropy-SGD algorithm that utilizes Stochastic Gradient Langevin Dynamics (SGLD) to estimate the local entropy gradient. This compares the marginal likelihood of two Bayesian priors centred at a solution with small and large local-entropic loss. In essence, the local-entropic loss was constructed by connecting the Gibbs distribution and optimization problems. A Bayesian prior may result in a small marginal likelihood when centred on an isolated solution instead of one centred within a wide valley (Dziugaite & Roy, 2017). This also confirmed that local extrema

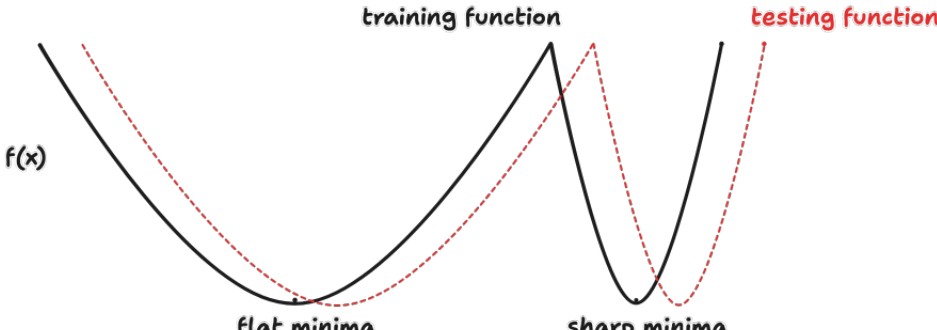

Figure 2: A sketch of valleys with flat minima vs. sharp minima. This illustrates the high sensitivity of a training function to perturbations, which negatively affects generalization (Keskar et al., 2016). Regions with flat minima are less complex and require low precision weights, making them likely to generalize better.

with low generalization have the majority of small eigenvalues in the Hessian. Despite attempts by Keskar et al. (2016) to remedy the sharp minima problems through data augmentation, conservative training and adversarial training, they could not overcome the problem.

The definition of flatness above does not entirely explain generalization due to the added complexity of deep models. For instance, in nonidentifiable rectified networks where some parameters in the model have little or no influence, there exist geometries resulting from symmetries created by these architectures that can be used to create comparable models with sharper minima (Dinh et al., 2017). In addition, directly associating Hessian-based sharpness to explain generalization is somewhat problematic because function reparameterization can drastically change and modify the parameters in the ReLU-MLP without changing the generalization properties (Wang et al., 2018).

Furthermore, sharpness, as defined in this section, cannot be utilized solely to determine the generalization behaviour of a deep neural network. It can be combined with the weight norm via PAC-Bayes (McAllester, 2003) to obtain suitable complexity measures. PAC-Bayes bounds are always valid or time uniform for fixed sample sizes and also hold for stopping times (Chugg et al., 2023). The authors argue that the perturbation loss and empirical loss gap can be used as a sharpness metric.

The smoothness of model generalization can also be associated empirically with the local characteristics of a solution using the PAC-Bayes framework. This hypothesis was tested by choosing an ideal perturbation algorithm to optimize the perturbed empirical loss for superior model generalization. Instead of using heuristic-based perturbation approaches (Khan et al., 2018; Jastrzębski et al., 2017; Zhu et al., 2018), Wang et al. (2018) included the second order bound information from Dziugaite & Roy (2017) for better PAC-Bayes bound optimization level. This technique also illustrated some inherent regularization properties by improving generalization performance on unseen data.

### 2.3 Implicit generalization and effective complexity

The independent variable in Figure 1 is model complexity, and in conventional statistics, such complexity is straightforwardly represented by the number of parameters in a model. An alternative way to reconcile traditional bias-variance arguments with evidence such as that of Zhang et al. (2021) is to employ alternative definitions of complexity. This has been a fertile avenue for recent thinking about generalization (Neyshabur et al., 2014). A detailed review is available in Hu et al. (2021), and we summarize some of the most important points here.

The principal insight of this approach is that various factors combine to determine the complexity of a classifier in practice. These include the number of parameters, the model architecture, the training algorithm and the training data employed. For example, variants of stochastic gradient descent (typically used for training

neural networks) tend to find solutions that are relatively close to the initial weight values. Thus, if these values can be shown to be of limited complexity by an appropriate measure, the existence of high-complexity networks elsewhere in weight space becomes irrelevant. If this is the case, the number of parameters is a misleading complexity measure. The inductive bias representing Occam's razor would still be relevant, but in this framework, it is no longer quantified by the model parameter count.

Several alternative complexity measures have been proposed, including measures based on norms (Neyshabur et al., 2015), sharpness in weight space (Keskar et al., 2016) and robustness to small perturbations (Xu & Mannor, 2012). However, none of these measures lead to a simple relationship between complexity and generalization (Neyshabur et al., 2017), similar to that in Figure 1.

### 2.4 Varieties of Interpolation

The concept of interpolation refers to a mathematical approach to predicting unknown values based on known observations (Davis, 1975). Research communities often define interpolation differently based on the techniques applied (Bishop & Nasrabadi, 2006; Aly & Dubois, 2005; Roy et al., 2013). In deep learning, various forms of interpolation have been utilized in recent years (Abir et al., 2021; Chahrour & Wells, 2022; Belkin et al., 2018; Adlam & Pennington, 2020). For example, the definition used by Balestriero et al. (2021) suggests that interpolation is assured if the sample of interest lies within the convex hull of the dataset. This definition is derived from low-dimensional intuitions and, as a result, experiences difficulties in high-dimensional spaces.

In contrast, our definition in Section 4.4 interpolates a sequence of elements between two points that least change the output. This approach does not depend on the dimensionality of the input and, therefore, does not rely on low-dimensional intuitions. Additionally, unlike Balestriero et al. (2021), our approach does not suffer from exponential data requirements. Our interpolation is exclusively implemented in input space and not in latent space. Similar work, generating images from the underlying network for embeddings, has been done using autoencoders (Singh & Ogunfunmi, 2021); our goal, however, is not to generate novel images but to understand how different network architectures generalize around training data points.

## 3 Benefits of depth

We investigate the effects of depth in standard deep neural networks, specifically deep convolutional neural networks and multilayer perceptrons. This is mainly informed by the research in Davel et al. (2020), which identified certain regularities in class-related activation patterns of ReLU-activated neural networks. This was observed in fully connected MLPs and provided helpful insight into the generalization abilities of DNNs. Davel et al. (2020) suggest that the cooperation of distinct classifiers formed by individual neurons results in redundancy and robustness, which increases if more network parameters are available. This contrasts with the classical bias-variance trade-off, discussed in Section 2, which posits that excess parameters beyond a certain number will decrease the accuracy of network generalization. Another regularity observed in feedforward neural networks of adequate depth was that class discrimination tends to shift to earlier layers. By mapping pixels into feature space, Zeiler & Fergus (2014) observed the hierarchical nature of CNN layers, where early layers respond to simple features and later ones to more complex features. These observations may provide an essential hint on the differences between CNNs and MLPs and thus will be central to our investigations.

The regularities observed in Davel et al. (2020) did not suggest a significant advantage in performance when the depths of MLPs are adjusted. For CNNs, in contrast, there are many reports that additional hidden layers produce improved generalization. As a basis for our further investigations, we first wish to establish a simple CNN architecture where these advantages of greater depth can be systematically observed. In subsequent sections, we strive to understand the causes of these improvements.

Section 3.1 describes the architectural design of the networks used to conduct the experiments. These design approaches are largely informed by the work presented in LeCun et al. (1998); Krizhevsky et al. (2012); Simonyan & Zisserman (2014); Zeiler & Fergus (2014); He et al. (2015). Section 3.2 explores the attributes

of MNIST, Fashion MNIST and CIFAR-10 datasets. Section 3.3 discusses the CNN performance on the abovementioned datasets at varying depth levels.

### 3.1 Experimental Setup and Generalization

We start by building a straightforward experimental setup to investigate the generalization behaviour of deep CNNs. We explore multiple architectures, with increasing depth, to conduct our training process. Figure 3 represents a network guided by the input dimensions of the datasets used, 28x28 for MNIST and FMNIST and 32x32 for CIFAR-10. Each feature extraction section of the network consists of a single convolution layer with a ReLU activation function and a pooling layer. To ensure that we keep the setup as simple as possible, we have omitted any explicit regularization, such as batch normalization and dropout. The number of filters per convolution layer, optimizer, learning rate and batch size were set as hyperparameters to optimize the learning process. We have also adopted small kernel sizes (3x3) with a unit stride as described by Simonyan & Zisserman (2014). Each feature map first passes through a (2x2) pooling layer to summarize the most prominent features and reduce dimensionality. All architectures used a single fully connected layer of 20 neurons to combine the flattened output of the feature extraction section of the network. The fully connected layer outputs were fed into the output layer. For its simplicity and probabilistic interpretation, a Softmax logistic function layer was used at the output to normalize the probability distribution for the predicted classes.

In addition, our setup included comparative MLP networks using a number of hidden layers of varying widths. Since there is no "standard recipe" for choosing these layer sizes, we experimented with various combinations of width and depth, and report on values that performed competitively. These contained either a total of 1024 hidden neurons, or 1024 neurons in the first hidden layer with a subsequent reduction in later hidden layers. The standard architecture without explicit regularization and a Softmax logistic function at the output was employed. All architectures were optimized to reduce a cross-entropy loss function for each training sample batch with varying sizes as hyperparameters.

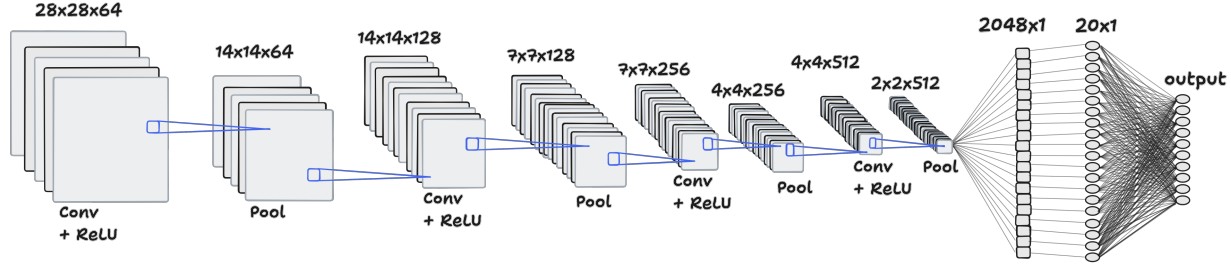

Figure 3: An architectural design of a deep CNN with average pooling and a single 20-neuron fully connected layer. The network depth of 4 layers was only constrained by the input dimensions. Note that the dimensionalities shown are for FMNIST and MNIST datasets and input dimensions change to 32x32 for CIFAR-10.

### 3.2 Data

Our experiments were conducted on the MNIST (LeCun et al., 1998), Fashion MNIST (Xiao et al., 2017) and CIFAR-10 (Krizhevsky et al., 2009) datasets. The datasets are standard, freely available and have widely been used as a point of reference for computer vision tasks in machine learning. Although these datasets are easy to recognize with state-of-the-art systems, they are sufficiently challenging to demonstrate the properties of the relatively simple CNNs that are our focus. This allows us to conduct experiments on relatively affordable hardware and prevents us from being distracted by many of the complications (such as data augmentation) required for state-of-the-art performance. Of course, this approach only makes sense if the lessons learned on these relatively simple systems and datasets also transfer to more competitive systems. We return to this issue in the concluding section.

MNIST is a dataset of real-world handwritten digits that range from 0 to 9. It has 70,000 grayscale image samples with 28x28 input features, of which 60,000 were used for training and 10,000 for testing. There are 10 distinct classes of almost equal size, as shown in Figure 4a. Each image pixel value was normalized and rescaled to a range between 0 and 1 to ensure that all input feature vectors have similar data distribution for faster network training. The Fashion MNIST dataset, shown in Figure 4b, is similar to MNIST in that it has the same number of samples, input features, and 10 distinct classes and is also in grayscale. Fashion MNIST was created as a more challenging option to the MNIST dataset. CIFAR-10 was the third dataset comprising real-world object colour image samples. It has 50,000 colour (RGB) image samples with 32x32x3 input features each for training and 10,000 for testing. There are also 10 distinct classes, each with precisely 5,000 colour image training samples, as shown in Figure 4c. The normalization process was the same for all the datasets; no data augmentation was used.

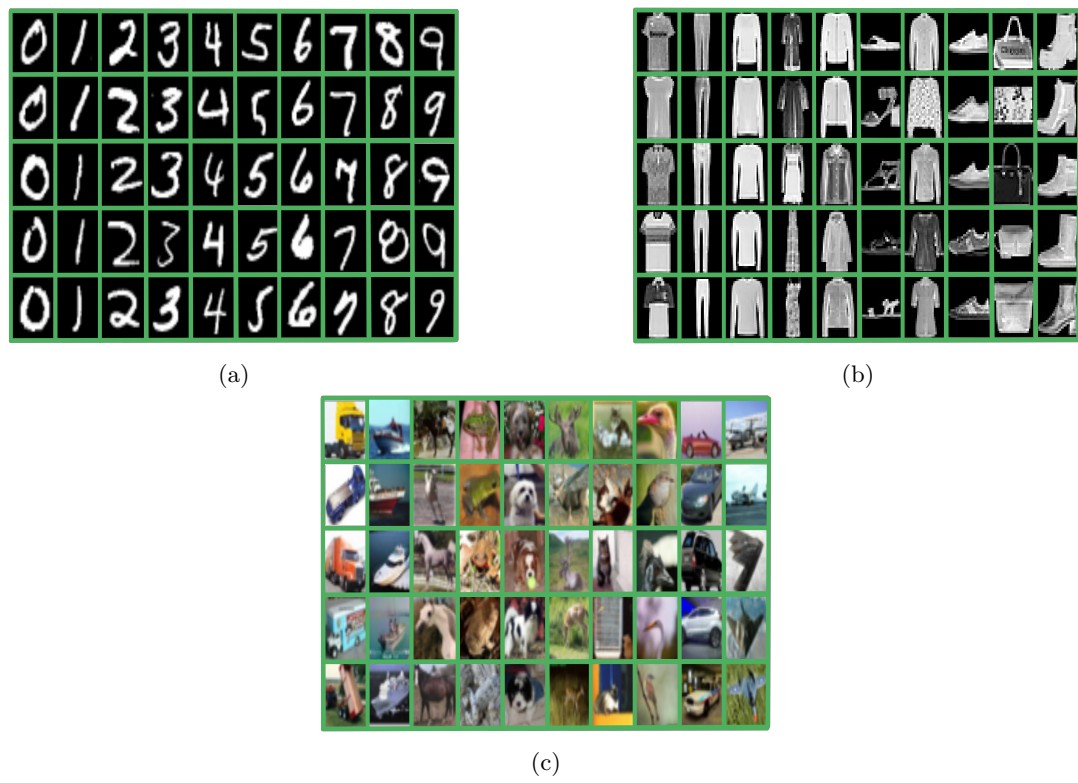

(a)

(b)

(c)

Figure 4: a) MNIST and b) Fashion MNIST single-channel and c) CIFAR-10 three-channel class dataset samples. Class members are grouped vertically for all 10 classes.

### 3.3 Results

We started our experiments with an initial convolutional layer of 64 neurons (filters). Following the standard CNN architecture design, all subsequent layers had double the number of neurons from the previous layer. It should be noted that the network was limited to a maximum depth of four due to the lack of dimensionality after the last pooling layer. The general architecture remained the same for all the datasets but required different hyperparameters. No explicit regulariser was used to train the CNN and MLP models. The models reached close to 0% training classification error but were saturated after several epochs. MNIST and FMNIST models showed similar validation error curves, as shown in Figures 8 and 9, respectively. The CIFAR-10 model output plots in Figure 10 show convergence around 50 epochs but display more distinct validation error curves between the network depths than MNIST and FMNIST.

The MLP networks were also trained on standard architectures with one to four layers and hidden-neuron configurations as described in Section 3.1. The models for all datasets were trained to 0% training classifi-

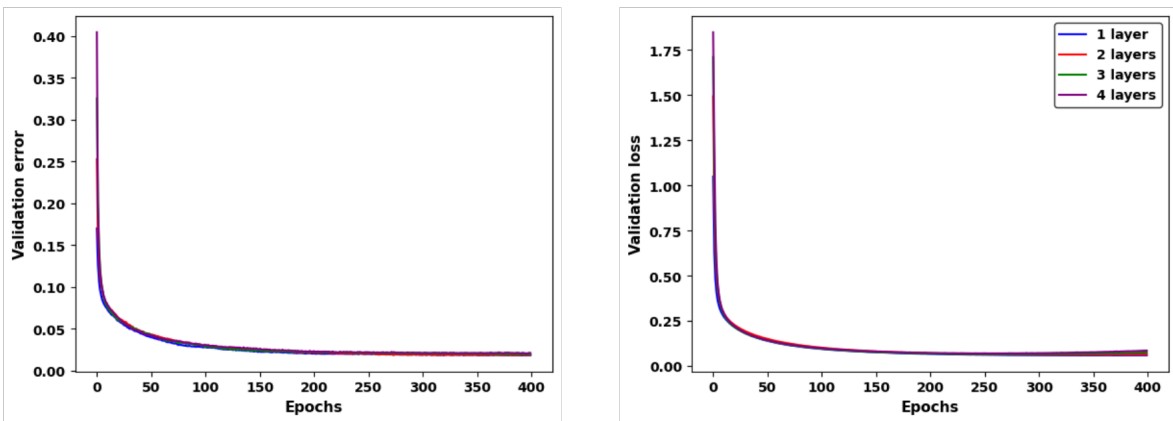

Figure 5: MLP validation error rate and loss per network depth for MNIST dataset.

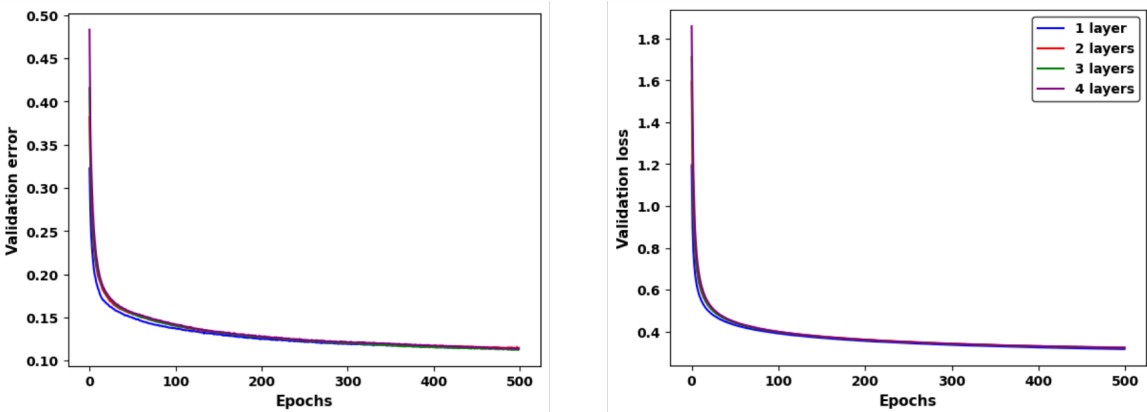

Figure 6: MLP validation error rate and loss per network depth for FMNIST dataset.

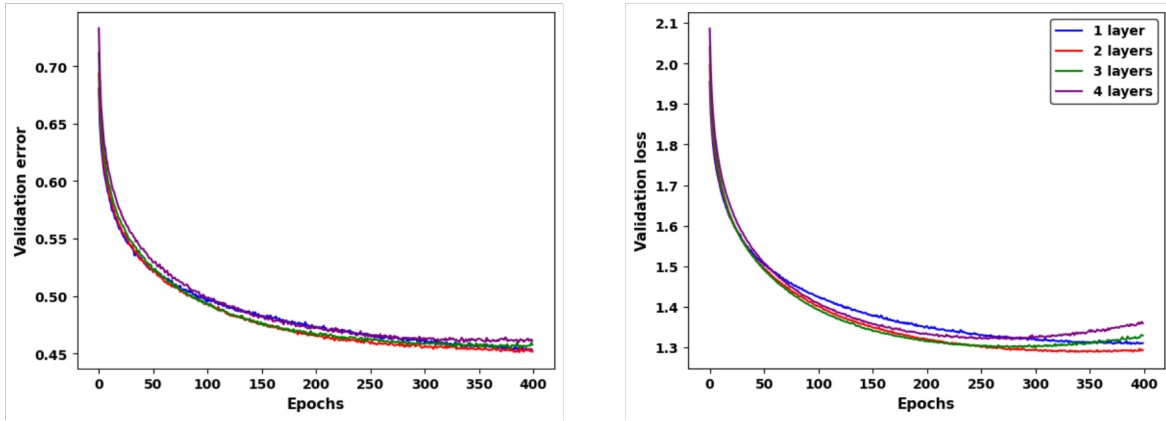

Figure 7: MLP validation error rate and loss per network depth for CIFAR-10 dataset.

cation error. We selected five trained models from random initial seeds for each network configuration from both MLPs and CNNs and calculated the standard error to show the consistency of the training process.

Once the training process converged, we observed a gradual decrease in validation loss for the MNIST dataset in CNNs. However, the validation loss curves for the FMNIST and CIFAR-10 datasets, with greater depths, show a U-shaped performance curve up to 100 epochs. Figure 10 displays a U-shape pattern but bears no

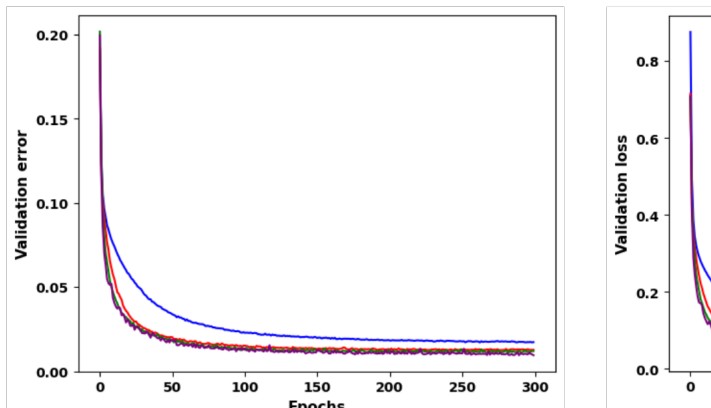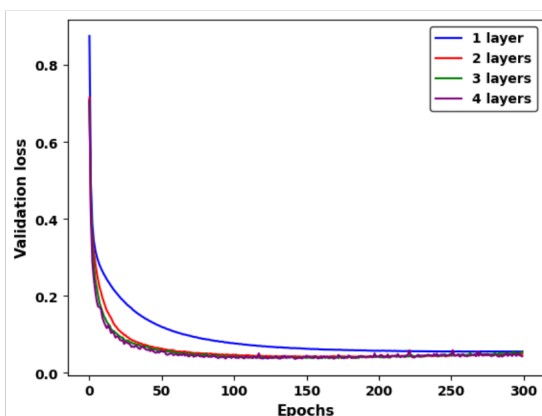

Figure 8: Validation error rate and loss per network depth for MNIST dataset. The CNN starts with 64 neurons in the first layer and doubles in subsequent layers.

evidence of a distinct risk peak and gradual decrease of the validation error after some threshold to fully realise the double descent phenomenon (Yang et al., 2020). Our observations also show that the deeper networks for the MNIST dataset have a lower validation loss than shallower ones. This is generally the opposite behaviour displayed by the FMNIST and CIFAR-10 networks. The MLP model results, shown in Figures 5, 6 and 7, exhibit a consistent trend in the validation error and loss for all datasets. The MLP networks perform similarly for all datasets, across different depths, which suggests that no benefit is gained for added depth.

Table 1 summarises the model performance for all datasets for the CNN architectures. The networks showed good validation error rates on unseen data as highlighted by the systematic decrease in validation error. Additional layers improved the model's generalization performance. Although there was some increase in validation loss for various depths for FMNIST and CIFAR-10 datasets, this did not impact the validation error rate. (Of course, early stopping was an option, but was deemed unnecessary based on the validation-error curves.) The deeper the network, the better the overall model validation error rate. This observation was consistent for similar architectures with fewer filters but with reduced accuracies.

Shallower networks for the CIFAR-10 dataset could not fit the training data due to insufficient capacity. As a result, networks with increased capacity produced the best results and met our requirement to train models to almost 0 % training error. This systematic benefit of network depth stands in sharp contrast to the behaviour of MLPs, where empirical evidence in Table 2 showed that shallow networks closely approximate the performance of optimal deep networks if the hidden layer sizes are chosen appropriately. In general, an increase in the number of layers saw an increase in the number of trainable parameters for CNNs and improved generalization, as opposed to MLP networks where generalization did not change systematically with depth. Hence, a single-layer MLP network with 1024 neurons will be used in subsequent experiments.

## 4 Piecewise linear model and an inductive bias

Our experiments have shown that adding more convolutional layers improves model generalization. This empirical evidence indicates that deeper networks perform better than their shallower counterparts. These networks seem to create more accurate classifiers and as we increase the number of layers, they seem to cope more successfully with distortions. CNNs, through the combination of convolution and pooling, appear to automatically create mechanisms that ensure that two things are judged as similar when they are visually similar. The more capacity the network has, the more flexible this similarity metric seems.

The empirical evidence provided in Section 3.3 confirms the conclusions of Zhang et al. (2021). This is in direct contrast to the classical learning theory and its key concept of bias-variance tradeoff, which would imply that overparameterized, high-dimensional and high-complexity networks should not perform well on

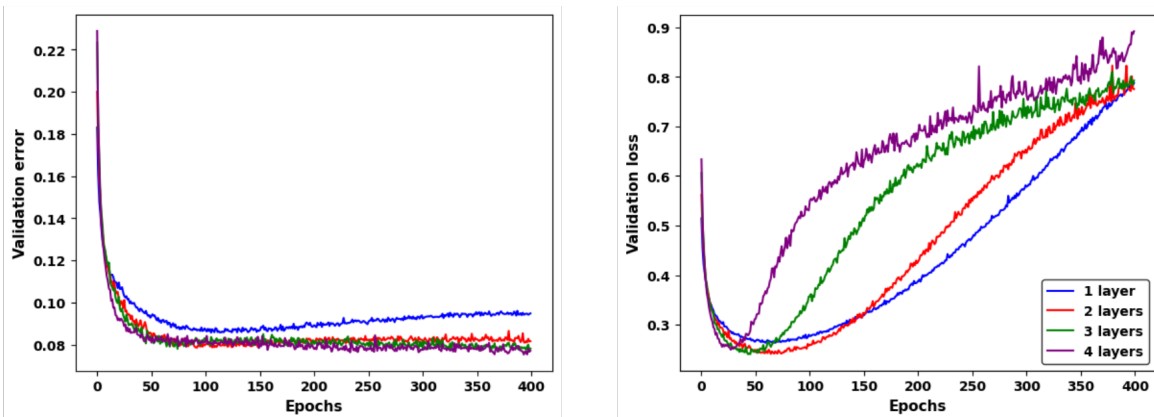

Figure 9: Validation error rate and loss per network depth for FMNIST dataset. The CNN starts with 64 neurons in the first layer and doubles in subsequent layers.

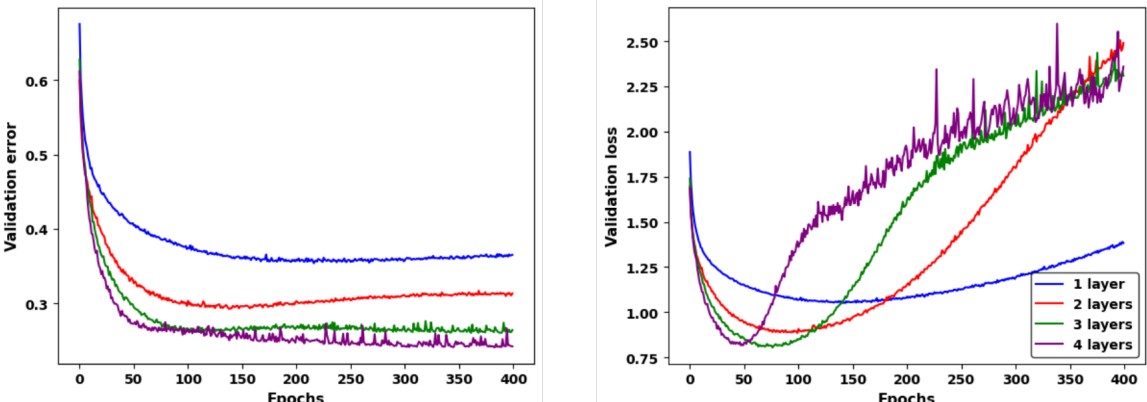

Figure 10: Validation error rate and loss per network depth for CIFAR-10 dataset. The CNN starts with 64 neurons in the first layer and doubles in subsequent layers.

unseen data. Despite the clear experimental evidence, there is currently no comprehensive explanation for the observed pattern of outperformance (which strongly favours deeper networks).

In Section 4.1, we take an alternative approach to SLT to understand the generalization capabilities of deep CNNs. We do this by introducing the perspective of classifiers as converters. This approach motivates us to focus on the locally linear mapping computed by deep CNNs to understand the properties of CNN-based converters in Section 4.2. In Section 4.3, we develop the precise analytic expression of the locally linear representation of a convolution layer. Section 4.4 investigates the linear interpolation between training samples from the same class by networks of varying depth, using the linear representation to probe the properties of such converters in order to gain insight into the benefits of greater network depth.

## 4.1 Converters and classifiers

We propose viewing classifiers as converters between the following two metric spaces.

  i The input or feature space, and

  ii Output space which reflects some measure of the likelihood that each of the target classes occurred (e.g. the class posterior probabilities)

Table 1: Average CNN performance results for increasing depth on the MNIST, FMNIST and CIFAR-10 validation sets. Each number in the filters column represents the number of neurons per subsequent convolutional layer. Note that the number of neurons in the fully connected layer in CNNs was set to 20 for all experiments. Average validation error and standard error as in Table 2.

| | | MNIST | | FMNIST | | CIFAR-10 | |
|---|---|---|---|---|---|---|---|
| Layers | Filters | Avg. error | Std error | Avg. error | Std error | Avg. error | Std error |
| 1 | 64 | 1.695 | 0.046 | 9.482 | 0.150 | 36.50 | 0.39 |
| 2 | 64+128 | 1.255 | 0.046 | 8.159 | 0.164 | 31.33 | 0.56 |
| 3 | 64+128+256 | 1.208 | 0.028 | 7.808 | 0.098 | 26.40 | 0.51 |
| 4 | 64+128+256+512 | **0.959** | 0.036 | **7.708** | 0.035 | **24.21** | 0.64 |

Table 2: Average MLP performance results for increasing depth on the MNIST, FMNIST and CIFAR-10 validation sets. Average validation error and standard error thereof computed over 5 trained models starting from different random initializations.

| | | MNIST | | FMNIST | | CIFAR-10 | |
|---|---|---|---|---|---|---|---|
| Layers | Filters | Avg. error | Std error | Avg. error | Std error | Avg. error | Std error |
| 1 | 1024 | 1.93 | 0.019 | 11.32 | 0.05 | 45.25 | 0.27 |
| 2 | 768+256 | 1.91 | 0.014 | 11.46 | 0.04 | **45.15** | 0.15 |
| 2 | 1024+512 | **1.82** | 0.013 | 11.40 | 0.06 | 45.63 | 0.12 |
| 3 | 640+256+128 | 1.93 | 0.029 | 11.29 | 0.05 | 45.76 | 0.12 |
| 3 | 1024+512+256 | 1.89 | 0.031 | 11.04 | 0.04 | 45.26 | 0.13 |
| 4 | 576+256+128+64 | 1.99 | 0.029 | 11.44 | 0.08 | 46.08 | 0.14 |
| 4 | 1024+512+256+128 | 1.91 | 0.038 | **10.93** | 0.07 | 45.59 | 0.17 |

Naive intuitions about metric spaces are heavily influenced by our everyday experience of the three spatial dimensions. Most prominently, these intuitions assume a near-Euclidean metric in a typical space after appropriate scaling. For the output space, that intuition is helpful in that one expects the outputs of a well-trained classifier to be concentrated on the unit vectors representing each of the classes in the standard one-hot encoding. Additionally, the Euclidean distance from any of these vectors reasonably indicates how unlikely the corresponding class is.

However, the input space behaves quite differently. Consider the binary images in Figure 11: if the brightness of each pixel is treated as a distinct dimension in the input space, the images in Figure 11b and Figure 11c are at the same Euclidean distance from the image in Figure 11a but in a metric that corresponds to the semantics of the image, they are quite different. Therefore, we can ask how a successful classifier transforms the highly distorted metric at the input to the more regular metric at the output.

This seems like a fruitful way to investigate CNNs, since such problems with severely distorted input metrics are precisely those for which these classifiers generalize with extraordinary accuracy. For more *regular* input spaces, such as those defined by conventional feature extraction, DNNs have not been shown to perform particularly well (Grinsztajn et al., 2022; Gorishniy et al., 2021).

One way to understand the relationship between the input and output metrics is to study their respective invariances under the semantic interpretation. If the output variables approximate posterior probabilities, their values will approximately sum to 1, and thus, for $C$ classes, there are $C - 1$ independent directions of variation and a single invariance. On the input side, matters are much more complicated. Consider, for example, an input image of size $D = N \times M$ pixels. If we were to perform a linear mapping from input

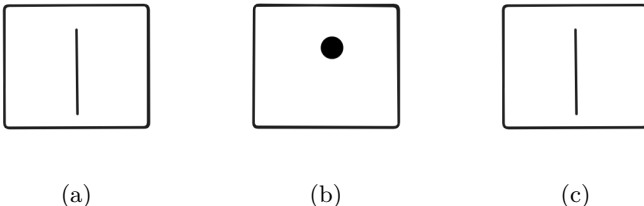

(a)                              (b)                              (c)

Figure 11: Image (c) is shifted one pixel to the right compared to image (a); hence, the respective Euclidean distances between images (b) and (c) to image (a) are the same.

to output, the $C - 1$ dimensions of class variation in the input space would generically map to a $C - 1$ dimensional subspace of the input space. Thus, there would be $D - C + 1$ dimensions, which are invariants for classification. The same relationships hold locally for a network with a piecewise-linear transfer function, such as an MLP or CNN with ReLU non-linearities. Within each linear patch of the piecewise-linear input space, there will be $D - C + 1$ class-invariant dimensions and $C - 1$ dimensions that span the class variations.

We have previously shown (see Section 3.3) that CNN architectures with different numbers of convolutional layers differ systematically in how well they generalize. We propose to study the source(s) of those differences by investigating these two local subspaces for different CNN architectures.

### 4.2 Input-output mapping in a CNN

We first express the convolution operation as a conventional matrix-vector product to find the local linear mapping computed by a CNN. Let us denote the activities of the nodes in layer $i$, channel $k$ of a CNN by the vector $\mathbf{x}_k^i$, where the rows of an image plane or feature map are concatenated to form a vector. These vectors can, in turn, be concatenated to form a vector $\mathbf{x}^i$ containing all the activities of layer $i$. As shown in Section 4.3 below, the convolution and pooling operations in one layer can be **locally** combined into a single linear operation

$$\mathbf{x}^{i+1} = \mathbf{W}^i \mathbf{x}^i, \tag{1}$$

where $\mathbf{W}^i$ is a matrix that depends on the convolution filters, the activations of the ReLUs, and the pooling function in layer $i$ (details below). The output of the final convolution layer is therefore

$$\mathbf{x}^{L-1} = \prod_{i=0}^{L-1} \mathbf{W}^i \mathbf{x}^0. \tag{2}$$

Any fully connected layers are similarly locally linear so that a single matrix-vector product can locally represent the mapping from input to logits. As will become clear below, we do not need to include the softmax operation in our representation, so our focus will be on the mapping

$$\mathbf{z} = \mathbf{W}(\mathbf{x}^0) \mathbf{x}^0, \tag{3}$$

where $\mathbf{z}$ is the vector of logits and $\mathbf{W}$ is a product of the matrices representing the convolution layers as well as the fully connected layers, which depend on the input $\mathbf{x}^0$ according to the derivation in Section **??**.

We aim to use this representation to probe the input-output mappings of CNNs with different levels of generalization. One way to do this is to visualize the neighbourhoods adjacent to representative training samples. We select a sample $\mathbf{x}^{0c}$ from class $c$, and then trace out the way the input changes as we move directly in the directions of each of the other classes $c'$. Since we know that the class-variable subspace contains $C - 1$ dimensions, these $C - 1$ variations form a complete linear basis for the local between-class variability of the input space. We compute these variations in detail using the following procedure.

1. Select two classes $c$ and $c'$, also, a representative sample $\mathbf{x} = \mathbf{x}^{0c}$ (i.e. a sample with high correct posterior probability in the trained classifier) from class $c$.

2. Compute the mapping matrix $\mathbf{W}(\mathbf{x})$ centred on $\mathbf{x}$. By construction, it yields the output logits $\mathbf{z} = \mathbf{W}\mathbf{x}$.

3. Move in a straight line between this sample and the closest location classified as $c'$. Since the rows of $\mathbf{W}(\mathbf{x})$ map from the input space to the class logits, this motion is

$$\mathbf{x} \to \mathbf{x} + \epsilon(\mathbf{W}_{c'.} - \mathbf{W}_{c.}), \tag{4}$$

where $\mathbf{W}_{i.}$ is the $i$-th row vector of $\mathbf{W}$.

4. Output this image and stop the process if the posterior of $c'$ exceeds that of $c$. Otherwise, repeat the process from Step 2.

This process produces images representing one class variability dimension around $\mathbf{x}^{0c}$. By repeating it for all possible choices of $c'$, we should be able to gain a reasonable intuition of the types of invariances captured by the network at that location, and further investigation for all choices of $c$ will enhance that intuition.

Alternatively, we can use a similar process to investigate the approximately invariant space between samples that belong to the same class. Such an investigation may suggest a metric of intra-class invariance that predicts how well a given network will generalize, as shown below.

## 4.3 Linear representation of a convolution layer

A minimal convolution block in a standard CNN consists of three major stages:

1. Each input plane $\mathbf{x}_k^i$ is convolved with the relevant kernel $\mathbf{f}_{kk'}^i$ ($i$ and $k$ are plane and spatial indices, respectively):

$$\mathbf{y}_k^i = \sum_{k'} \mathbf{x}_{k'}^i \mathbf{f}_{kk'}^i. \tag{5}$$

2. The results of the convolution are passed through a ReLU non-linearity:

$$z_{k,ab}^i = \max(y_{k,ab}^i, 0), \tag{6}$$

where the indices $a, b$ refer to the coordinates of the feature map.

3. Finally, the inputs of the next layer are computed by pooling the relevant $z$ values with an appropriate operator, either maximum or average pooling.

$$\mathbf{x}_k^{i+1} = \text{Pool}(\mathbf{z}^i). \tag{7}$$

To implement the convolution as a matrix-vector product, we concatenate the images or feature maps on all channels into a single vector, as described above. The convolution is achieved by inserting the relevant kernel value for each product in the corresponding matrix position, for example, row $r = k \times D^i + d$ produces the output at position $d$ of the $k$-th channel and therefore consists of the concatenation of the elements of $\mathbf{f}_{kk'}^i$, offset by $d$ positions and correctly spaced. This matrix we denote by $\mathbf{V}^i$. For a 5x5 input, 3x3 kernel, two input channels and two output channels, for example, the structure of $V^i$ is as follows, assuming zero padding and same convolution (where the dimensions of the input and output are equal).

$$\begin{bmatrix} f_{00,11}^i & f_{00,12}^i & 0 & 0 & 0 & f_{00,21}^i & f_{00,22}^i & 0 & \cdots & 0 & f_{01,11}^i & f_{01,12}^i & \cdots \\ f_{00,10}^i & f_{00,11}^i & f_{00,12}^i & 0 & 0 & f_{00,20}^i & f_{00,21}^i & f_{00,22}^i & 0 & \cdots & f_{01,10}^i & f_{01,11}^i & f_{01,12}^i & \cdots \\ \vdots & & & & & & & & & & & & \\ f_{00,01}^i & f_{00,02}^i & 0 & 0 & 0 & f_{00,11}^i & f_{00,12}^i & 0 & \cdots & 0 & f_{01,01}^i & f_{01,02}^i & \cdots \\ f_{00,11}^i & f_{00,12}^i & 0 & 0 & 0 & f_{00,21}^i & f_{00,22}^i & 0 & \cdots & 0 & f_{01,11}^i & f_{01,12}^i & \cdots \\ f_{00,10}^i & f_{00,11}^i & f_{00,12}^i & 0 & 0 & f_{00,20}^i & f_{00,21}^i & f_{00,22}^i & 0 & \cdots & f_{01,10}^i & f_{01,11}^i & f_{01,12}^i & \cdots \\ \vdots & & & & & & & & & & & & \end{bmatrix}$$

The ReLU function replaces all negative entries in $\mathbf{V}^i\mathbf{x}^i$ with 0 and this operation can therefore be implemented locally by replacing the rows of $\mathbf{V}^i$ corresponding to zero entries in this product with the zero vector, producing the matrix $\mathbf{V'}^i$.

Finally, pooling is implemented by carrying out the corresponding operation on the rows of $\mathbf{V'}^i$. For max pooling, only those rows corresponding to the chosen maxima in each pool are copied to $\mathbf{W}^i$, or for average pooling, the rows corresponding to the pooled values are averaged together to form the rows of $\mathbf{W}^i$.

This construction allows us to create a matrix $\mathbf{W}^i$ which locally maps between the input and output planes of a convolutional block, and by multiplying these matrices together, we compute the locally linear map between the input and output of a CNN.

### 4.4 Interpolation between training samples

The linear mapping between input and output is only valid in a region where all nonlinear elements remain in the same state. That means all negatively saturated ReLU units stay negatively saturated, all ReLU units in the linear region remain so, and all max-pooling operations select the same argument. Any investigation of network behaviour must deal with switches between different linear domains. We wish to interpolate between neighbouring training samples to understand how different CNNs generalize away from the training samples. This process requires us to track how the linear domains change as we move between those samples. Our initial investigations in this regard revealed two challenges.

   i The max-pool arguments switch rapidly, which requires investigating a large number of linear domains, even for small changes in the input.

   ii Linear interpolation between input samples creates nonsensical interpolated images. This should have been expected, given the highly non-Euclidean nature of the input space.

Two mechanisms can be invoked to address these issues. On the one hand, we have shown that the benefits of CNN depth are equally clear in networks with *average pooling* rather than max pooling, even if the absolute performance is somewhat better with max pooling. Thus, we can address the first issue by modifying the above algorithm to utilize average pooling.

The second proposal is to develop a modified interpolation scheme that accounts for the properties of input space. Let us consider interpolation between two inputs $\mathbf{x}$ and $\mathbf{y}$ and let us restrict our attention to binary images to simplify the calculation. These images differ in $d$ positions and interpolation in $n$ steps requires that these $d$ different values be systematically inverted. We therefore propose that, on each successive step of interpolation, $d/n$ of the image locations with differing values be changed.

The value of the linear representation is now apparent, since we can use it to efficiently choose those $d/n$ locations to minimize the change in the output values or logits, thus implementing a suitable form of *interpolation*. To minimize the change in the logits, we want to choose the first interpolated image $x_1$ to minimize

$$\|\mathbf{W}(\mathbf{x})(\mathbf{x}-\mathbf{x_1})\| = (\mathbf{\Delta_1^T}\mathbf{W}(\mathbf{x})^\mathbf{T})(\mathbf{W}(\mathbf{x})\mathbf{\Delta_1}),$$

where $\Delta_1$ is the change vector $\mathbf{x}-\mathbf{x_1}$. Since $\mathbf{W}(\mathbf{x})$ can be calculated with the process described in Section 4.4, this minimization with respect to $\Delta_1$ is a simple matter in principle. In practice, though, it may be necessary to do so by steepest descent to limit computational cost.

We can use this process to generate a sequence of images $\mathbf{x}, \mathbf{x_1}, \mathbf{x_2}, ..., \mathbf{x_n} = \mathbf{y}$, producing similar outputs and interpolating between the original images. As mentioned above, this should give us significant intuition about the invariances representing the network in the neighbourhood of $\mathbf{x}$ and $\mathbf{y}$, and thus to understand the inductive bias that leads to improved performance with deeper networks.

## 5 Results: image interpolation

Our application of the concepts introduced in the previous subsection begins with interpolating two random binary images from the same class. In this context, a binary image was created by assigning each pixel value

greater than zero to one and leaving the zero pixels unchanged. Figure 12 shows a pair of samples from each class in the MNIST dataset, with standard images to the right and binary versions to the left. The algorithm in Section 4.4 calls for a simple pixel-by-pixel comparison between pairs of images to determine the modified pixels. These pixels are used to calculate the amount of change for each successive interpolation step. For example, the first and second binary samples from class zero in Figure 12 have 106 mismatched pixels. If we opt to interpolate in 10 steps, we can change the following number of pixels in sequence: [ 11, 11, 11, 11, 11, 11, 10, 10, 10, 10 ]. The choice of which bit to flip was based on which change affected the output logits the least. This process was carried out for CNNs of different depths and a single-layer MLP network of adequate capacity, and typical examples of the results are shown in Figure 13.

The interpolation plots show that the deeper the network, the smoother the interpolated images compared to shallower ones. This means that bits change in a way to maintain image integrity so that the digits are kept intact for deeper networks. This is most noticeable in cases where the two anchor images are substantially different from one another, as in Figure 14: all the networks create significantly distorted interpolated images, but the interpolations by the 4-layer network are (for example) more image-like than those of the 1-layer network or MLP.

There is a particularly significant difference between networks with 0, 1 and 2 convolutional layers, on the one hand, and those with 3 and 4 layers – networks with 4 layers behave much the same as those with 3 layers. This was also observed in our validation accuracy results and may result from the rather simple MNIST task; we expect the same trend to continue to deeper networks with higher-resolution images and more complex classification tasks. In addition, as we add more convolutional layers, the networks seem to generalize more appropriately. Thus, the most important inductive bias is suggested by the structure of the digits between the interpolation steps. This reasoning comes from the fact that all these networks of different depths were trained close to 0% error rate and had similar capacities. From these observations, it would be reasonable to suggest that something inherent or natural in the deeper CNNs leads to this natural-looking interpolation. Our concluding discussion explores this capability of deeper networks in more detail.

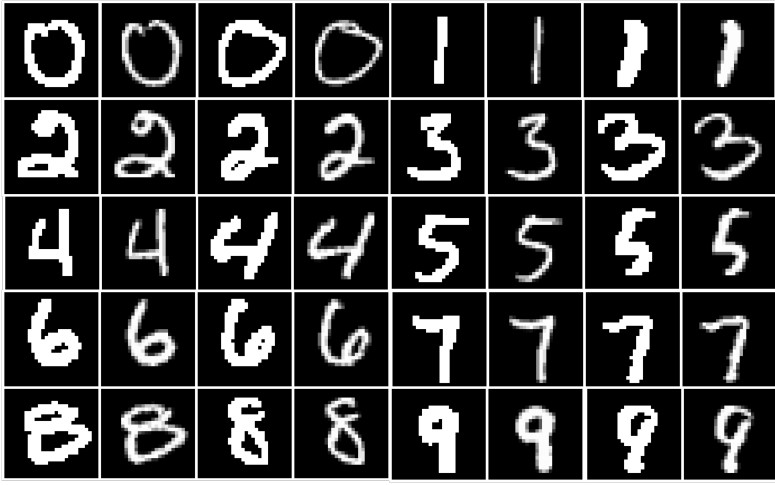

Figure 12: MNIST standard and binary samples. Samples on the left represent binary images with only 0s and 1s. Note that all pixel values greater than 0 were assigned to 1.

Based on our intuition above, a CNN with more convolutional layers has more reasonable interpolation. A reasonable image (if one interpolates between two intra-class images, for example, 4s, then the intermediate images must be 4-like) is fairly smooth and intact, as shown from Figure 13 and Figure 14. Similar reasoning should apply for inter-class images or images from different classes, as shown with the interpolation from class 4 to class 8 in Figure 15. Each intermediate image must be smooth to generalize well for all the randomly selected samples. This implies that the deeper the network (more hidden layers), the smoother and more intact the intermediate images. Intermediate images in deeper networks, therefore, have more low-frequency components than shallower networks, and shallower networks have more low-frequency components, isolated

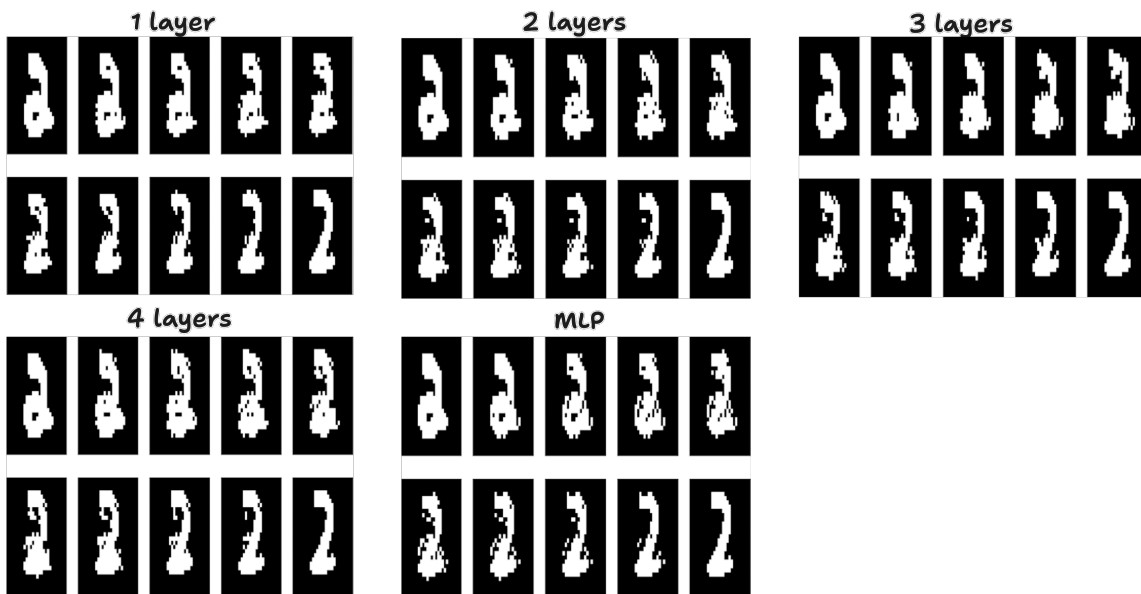

Figure 13: MNIST class 2 intra-class interpolation for the different CNN depths and an MLP network. Note that the layer numbers refer to the convolution layers in a given network, refer to Figure 3.

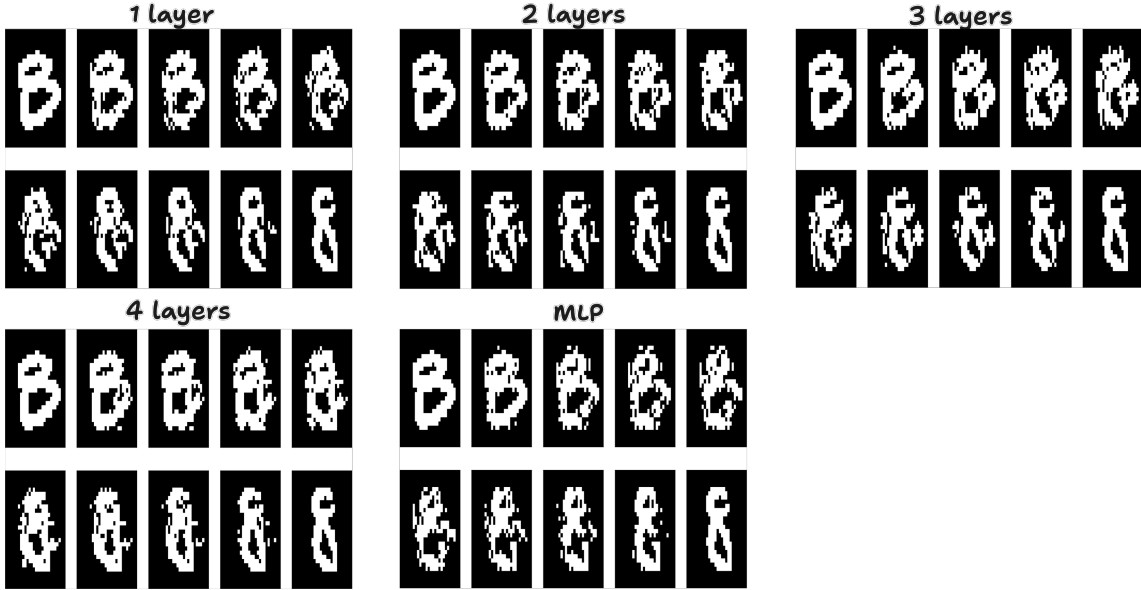

Figure 14: MNIST class 8 intra-class interpolation for the different CNN depths and an MLP network. Note that the layer numbers refer to the convolution layers in a given network, refer to Figure 3.

pixels and more random-looking pixels. We will now introduce a numerical measure aimed at capturing this intuition.

## 5.1 Bit change summations

One can imagine various sensible metrics to measure the difference between the intermediate images for the layers shown in Figure 13. These metrics should consider both the number of active pixels and the placement of those pixels. One obvious measure is the 2D FFT, which analyzes the frequency spectrum of

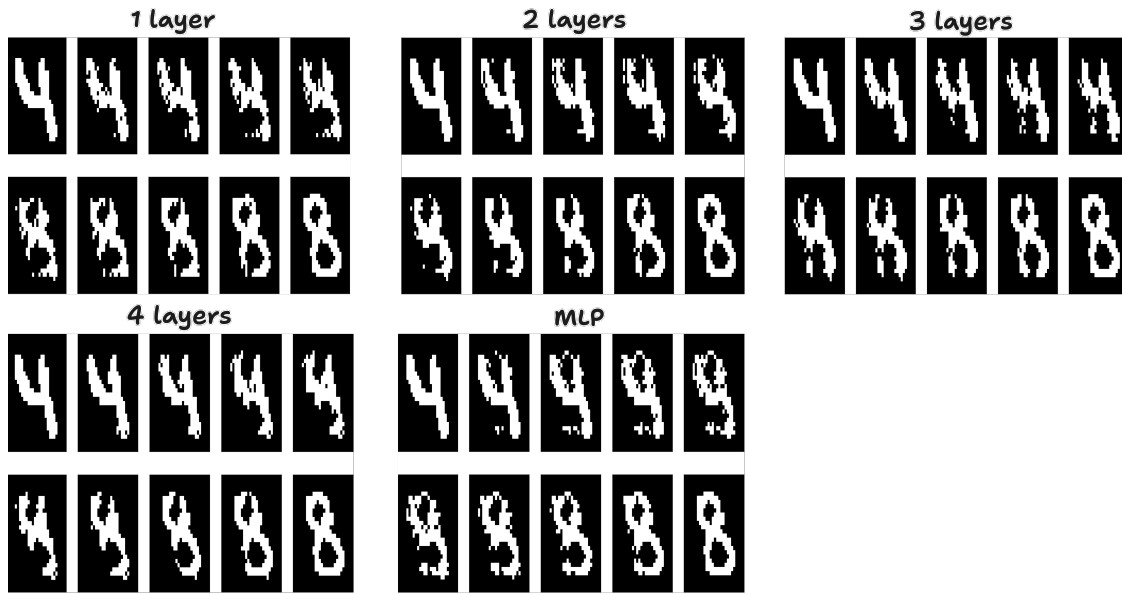

Figure 15: MNIST classes 4 and 8 inter-class interpolation for the different CNN depths and an MLP network. Note that the layer numbers refer to the convolution layers in a given network, refer to Figure 3.

the 2D signal or matrix. However, FFT-based smoothness measures can be problematic because they suffer from anisotropy and behave differently in coordinate directions than in rotated directions. In addition, the frequency cut-off that defines low-frequency components is a free parameter with appropriate values likely to be problem-specific. Therefore, we explored a simple alternative smoothness measure that was easier to substantiate and not dependent on fine-tuning.

Based on the differences observed in Figure 13, we wish to define a measure that identifies images that do not contain "unnatural" pixels. To this end, we determine whether every pixel is surrounded by other pixels with similar characteristics since a smooth binary image is one where most black pixels are surrounded by other black pixels and similarly for white pixels. Moreover, a real image with a good prior (one that generalized well) should be more connected and continuous, while a bad prior (one that did not generalize well) should introduce random-looking bits and pieces. So, to compute the smoothness measure, the binary image was scanned pixel by pixel from left to right to determine how many of each pixel's horizontal neighbours differed from the base pixel. The scanning process counted the number of times white pixels changed to black and black pixels to white. The exact process was repeated by scanning the pixels from top to bottom and counting the vertical changes. Both counts were added to give the number of un-surrounded edges a binary image had. This was done for each interpolated binary image, for $N$ iterations, for different CNN depths and a single-layer large-capacity MLP network.

Figure 16 shows the class-level bit counts for various CNN depths and the MLP network for the MNIST dataset. The extreme points represent the start and end average bit change counts for randomly selected binary images. A perfect interpolation would result in a straight line between the extreme points. Based on the observation, the MLP network seems to have the most bit switches on average for every class and thus generates intermediate images that are the least image-like. CNNs, on the other hand, performed better on average for all the classes and showed some improvement as more hidden layers were introduced. A clear performance distinction exists between a CNN with 1 hidden layer and 2 or more hidden layers. In addition, a network with 3 hidden layers seems to perform very similarly to the one with 4 hidden layers, although the latter performs the best on average. These relationships agree well with the model performance results in Section 3. Similar observations were made for the FMNIST dataset in Figure 17. In this case, the superiority of the deeper networks is more substantial, as evidenced by the lower average bit change counts for the interpolated binary samples.

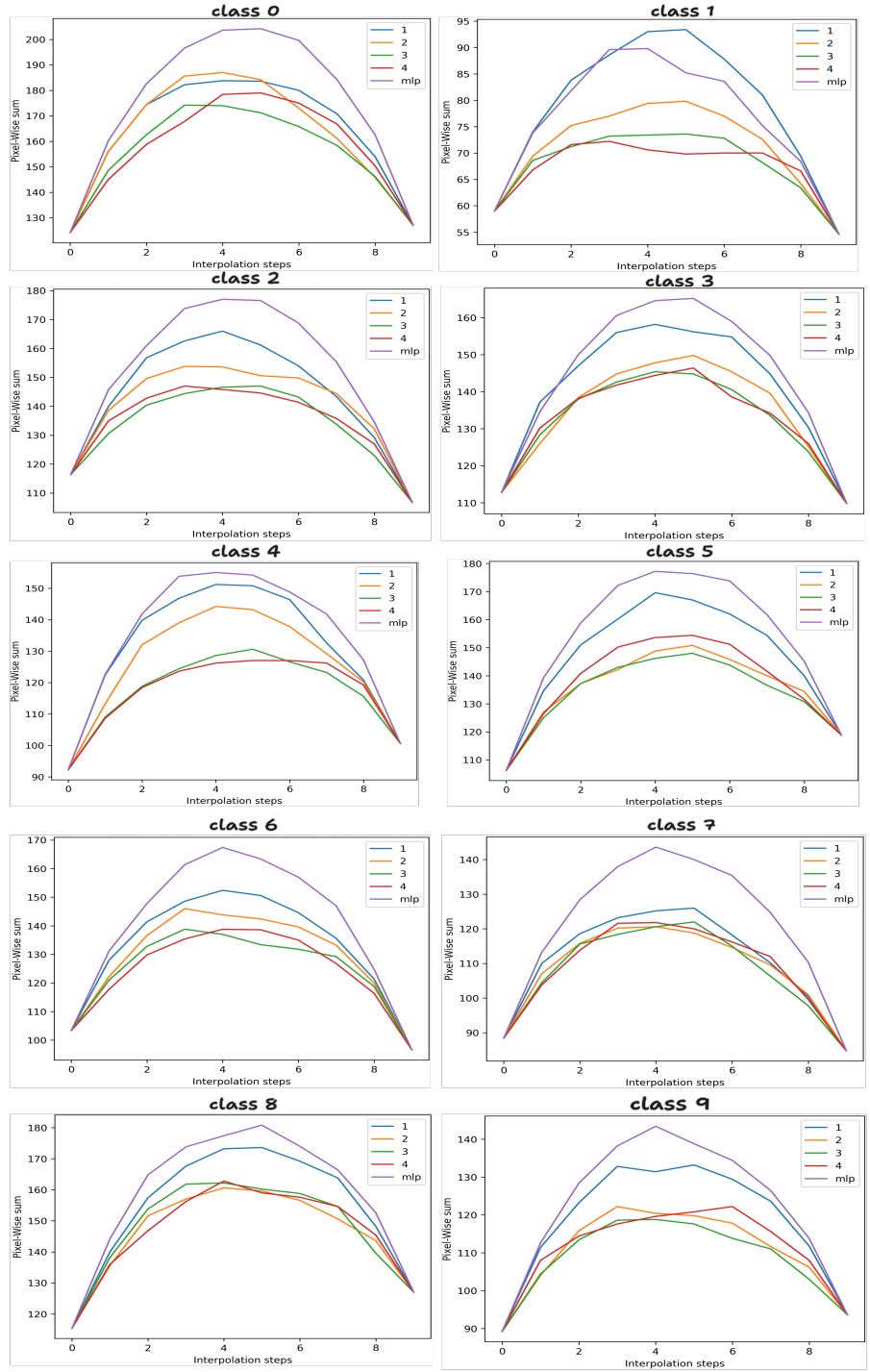

Figure 16: MNIST inter class bit counts for 4 CNNs and MLP network. The numbers within the legend represent the depth of each CNN and the MLP represents a single-layer, large-capacity network. Each interpolation point is computed by adding vertical and horizontal bit changes.

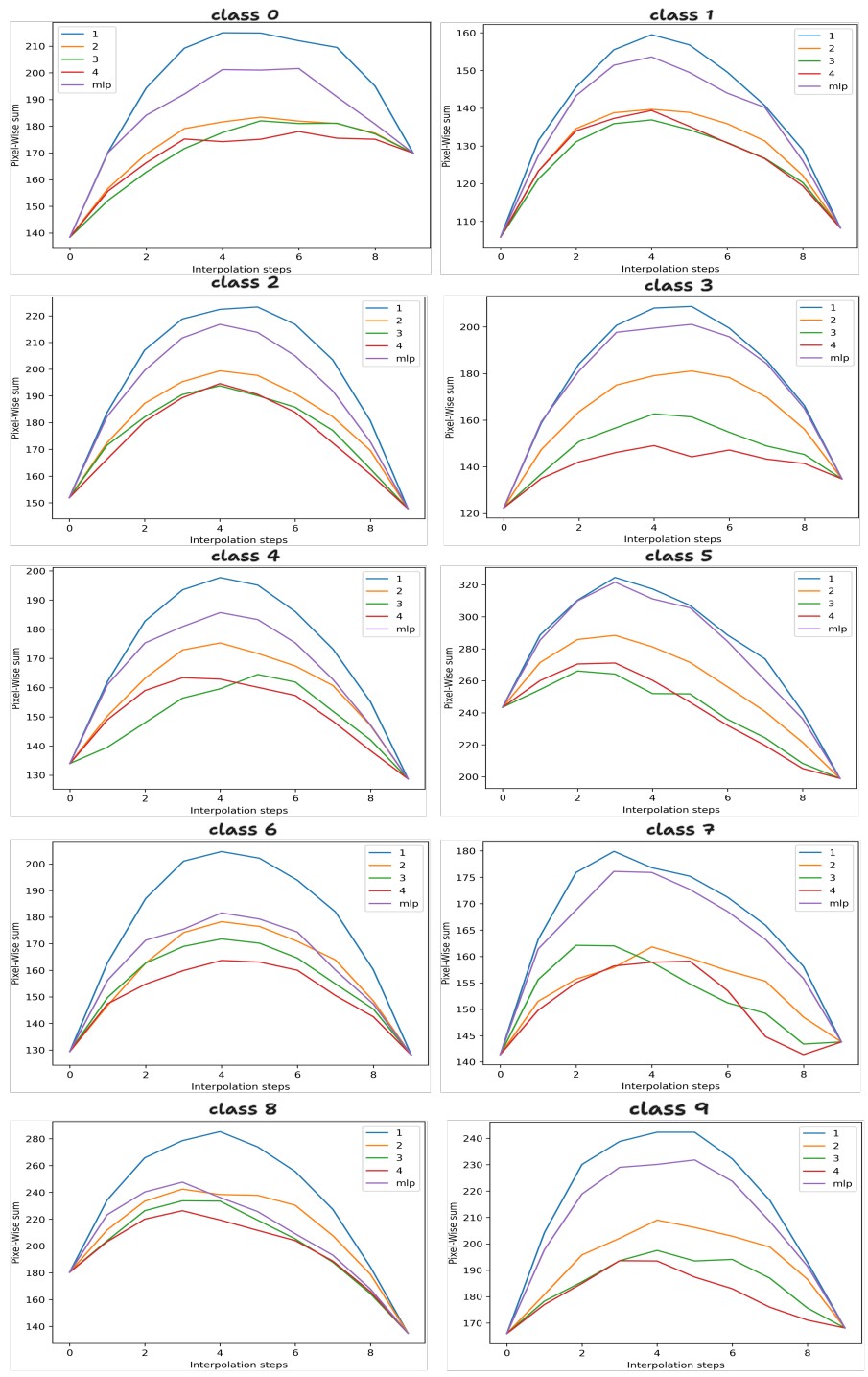

Figure 17: FMNIST inter class bit counts for 4 CNNs and MLP network. The numbers within the legend represent the depth of each CNN and the MLP represents a single-layer, large-capacity network. Each interpolation point is computed by adding vertical and horizontal bit changes.

# 6 Discussion

We have presented a framework that viewed classifiers as converters or maps between the feature space and output space to understand the generalization capabilities of CNNs. We developed a precise theoretical expression to describe the piecewise linear representation of a convolutional layer. In addition, we investigated the properties of such converters using linear interpolation between in-class training samples, applying the linear representation to probe their properties.

Our experiments showed that the maps for CNNs are biased to be increasingly smooth with greater depth, according to two different metrics. Since such smoothness is a characteristic of real images, deeper CNNs tend to outperform shallow CNNs and MLPs. By mapping pixels into feature space, Zeiler & Fergus (2014) observed the hierarchical nature of CNN layers, where early layers respond to simple features and later ones to more complex features. These observations may represent the mechanism whereby deeper networks have a more image-like bias since they can prefer ever-larger patches of realistically smooth input structure and give us confidence that our observations will also generalize to the deeper CNNs and more difficult tasks that represent the current state of the art.

In SLT, the inductive bias responsible for superior generalization is identified with adequate simplicity according to Occam's razor. Our results indicate that this is only one aspect of an appropriate bias – and for Neural Networks, not the most pertinent aspect. For CNNs, we find a quantifiable bias towards smoother images for deeper networks, and we hypothesize that this bias is the principal explanation for the excellent generalization achieved with deep CNNs. As networks and training data sets grow in size, this type of bias becomes dominant in generalization, and the simplicity bias (which is crucial for smaller networks) fades into irrelevance.

Finally, if our hypothesis is correct, it is likely that it applies (suitably modified) to transformer networks as well, given that such networks are reported to be similarly immune to overfitting. It will be interesting to investigate this issue in future work.

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

## A    Appendix

Table 3: The number of trainable parameters for various CNN depths. Each number in the 'filters' column represents the number of neurons per subsequent convolutional layer. Note that the number of neurons in the fully connected layer in CNNs was set to 20 for all experiments.

| CNN layers | Filters | Trainable params |
|:---:|:---:|:---:|
| 1 | 64 | 251,750 |
| 2 | 64+128 | 200,166 |
| 3 | 64+128+256 | 451,814 |
| 4 | 64+128+256+512 | 1,591,014 |

Table 4: The number of trainable parameters for various MLP depths.

| MLP layers | Hidden neurons | Trainable params |
|:---:|:---:|:---:|
| 1 | 1024 | 814,090 |
| 2 | 768+256 | 802,314 |
| 2 | 1024+512 | 1,333,770 |
| 3 | 640+256+128 | 700,682 |
| 3 | 1024+512+256 | 1,462,538 |
| 4 | 576+256+128+64 | 641,674 |
| 4 | 1024+512+256+128 | 1,494,154 |

