# OpenReview forum: "Network Depth and Inductive Bias in Convolutional Neural Networks"
_TMLR — Rejected by TMLR_

### Review · Reviewer_CqyA · 2025-02-16

**Summary Of Contributions:**

This paper explores why deeper CNNs outperform shallower architectures and MLPs in generalization. It challenges traditional statistical learning theory, particularly the bias-variance trade-off, and identifies smoothness as a key inductive bias unique to deeper CNNs. Through a naive experimental framework, mathematical modeling, and interpolation studies on datasets like MNIST and CIFAR-10, it demonstrates that depth enables CNNs to create smoother, more robust representations.

**Audience:**

No

**Broader Impact Concerns:**

Not applicable (no need any impact statement for ethical implications).

**Claims And Evidence:**

No

**Requested Changes:**

It is difficult to envision a path for this paper to meet the standards for acceptance by TMLR without a fundamental rethinking of its overall objectives. Both the central research question and the experimental design appear overly simplistic and lack the depth or sophistication expected in a submission of this caliber. A significant redesign of the study's focus and methodology is necessary to align with the expectations of the field.

**Strengths And Weaknesses:**

# Strengths:
I could barely find any concrete strengths of this paper, unfortunately.

# Weakness:

## 1. Figure Redundancy and Lack of Clarity:

Fig. 4 appears redundant: The inclusion of Fig. 4 does not provide any additional insights that aren't already evident from the context, making its presence unnecessary and adding to visual clutter in the paper.
Fig. 5 and Fig. 6 lack clarity: The curves in these figures are indistinguishable, making it difficult for readers to derive meaningful comparisons or conclusions. The figures need better differentiation, such as using clearer color schemes, line styles, or annotations to improve interoperability.

## 2. Poor Paper Structure and Writing:

Difficult to identify key conclusions: The paper is poorly structured and written, making it challenging for readers to quickly grasp its core contributions and findings. For example, sub-section 3.3 is generically titled "Results," offering no indication of its specific content. Furthermore, the lack of formatting, such as bold or underlined text to emphasize key points, exacerbates the difficulty in understanding the conclusions. A more structured and reader-friendly approach is necessary to enhance accessibility.

## 3. Unjustified Claims:

Conclusions about MLP performance lack justification: The statement, "The MLP networks perform similarly for all datasets, across different depths, which suggests that no benefit is gained for added depth," is unsubstantiated and overly broad. It fails to address several key considerations:
Whether appropriate regularization techniques were applied.
Whether better initialization methods, such as orthogonal initialization to preserve isometry, were used.
Whether architectural enhancements like skip connections were explored to potentially improve MLP performance.
Without addressing these factors, this conclusion either reflects a very narrow experimental scope that is not generalizable or fails to provide sufficient investigation of diverse settings. The authors should acknowledge the limitations of their approach and explore these aspects further to strengthen their analysis.

## 4. Lack of Novelty in Experiments:

Experiments lack innovation: The experiments on MLPs are not novel and add little value to the existing body of research. The findings about the limited benefits of added depth in MLPs have already been extensively explored in prior literature, and this study does not introduce any significant new insights or methodological advancements to build upon these observations.

---

> ### Author Response · Authors · 2025-03-13
>
> The reviewer is concerned that the simplicity of our approach may prevent ours from being a useful contribution. We hope that we can restructure the presentation (based on the feedback from all three reviewers) to overcome that concern, by focusing on those aspects of the work which do contain some novel insights on generalization in CNNs. In particular, we will address the concerns of the reviewer as follows.
>
> - _Fig. 4 appears redundant: The inclusion of Fig. 4 does not provide any additional insights that aren't already evident from the context, making its presence unnecessary and adding to visual clutter in the paper._
>
> The figure will be deleted.
>
> - _Fig. 5 and Fig. 6 lack clarity: The curves in these figures are indistinguishable, making it difficult for readers to derive meaningful comparisons or conclusions. The figures need better differentiation, such as using clearer color schemes, line styles, or annotations to improve interoperability._
>
> We agree that these two figures convey limited information. Based on feedback from all reviewers, this section will be moved to an appendix. We will see whether logarithmic scaling improves the visual separation, and if not we will replace the figures with a brief textual description of its main conclusions.
>
> - _Difficult to identify key conclusions: The paper is poorly structured and written, making it challenging for readers to quickly grasp its core contributions and findings. For example, sub-section 3.3 is generically titled "Results," offering no indication of its specific content. Furthermore, the lack of formatting, such as bold or underlined text to emphasize key points, exacerbates the difficulty in understanding the conclusions. A more structured and reader-friendly approach is necessary to enhance accessibility._
>
> Sections not necessary for our core argument will be deleted or moved to appendices, and headings of sections will be rephrased to be more specific. (For example, Sec 3.3 will be titled “Relationship between network depth and generalization”). We will also follow the suggestion to add more readable formatting in order to make our key points stand out clearly.
>
> - _Conclusions about MLP performance lack justification: The statement, "The MLP networks perform similarly for all datasets, across different depths, which suggests that no benefit is gained for added depth," is unsubstantiated and overly broad.It fails to address several key considerations: Whether appropriate regularization techniques were applied. Whether better initialization methods, such as orthogonal initialization to preserve isometry, were used. Whether architectural enhancements like skip connections were explored to potentially improve MLP performance. Without addressing these factors, this conclusion either reflects a very narrow experimental scope that is not generalizable or fails to provide sufficient investigation of diverse settings. The authors should acknowledge the limitations of their approach and explore these aspects further to strengthen their analysis._
>
> We will acknowledge these limitations, along with the informal observation that we have not been able to find any other architectural features which do, in fact, lead to significant MLP improvements with greater depth.
>
> - _Experiments lack innovation: The experiments on MLPs are not novel and add little value to the existing body of research. The findings about the limited benefits of added depth in MLPs have already been extensively explored in prior literature,and this study does not introduce any significant new insights or methodological advancements to build upon these observations._
>
> We will clarify that these experiments were intended as a baseline for the CNN investigations, which are the focus of our work.

---

### Review · Reviewer_PVKq · 2025-02-20

**Summary Of Contributions:**

This paper investigates potential reasons for CNNs exhibiting better generalization with increasing depth and capacity, which is inconsistent with what a bias-variance tradeoff would predict for basic notions of model complexity.  First, it establishes this behavior experimentally using MNIST, Fashion-MNIST and CIFAR10 and different model depths.  Then, it develops an interpolation method that illustrates the behavior of different depth networks' mapping functions in terms of measured smoothness and anecdotal appearance on binary datasets MNIST and Fashion-MNIST, finding that minimum-change paths from one point to another are indeed smoother and anecdotally more image-like for deeper networks.

**Audience:**

No

**Claims And Evidence:**

No

**Requested Changes:**

See above.  Most critical is further developing and addressing open some of the open questions in the interpolation section as mentioned in my review.  I also think the paper should be more focused around this part of the work.

**Strengths And Weaknesses:**

Overall, while there are a few interesting constructions in this work, I was not convinced by any of the arguments presented.  It also lacks focus, essentially having two parts, the first experimental val measurements and the second the interpolation method.  I don't think the first part adds much here, since these behaviors are already established including in some of the works already cited.

In my opinion the second part describing the interpolation technique is more novel.  Most current interpolation experiments interpolate linearly in the hidden dimension and decode or backproject, while this method chains based on minimizing classification difference between samples.  There does seem to be something in here, but still under-developed as I wasn't convinced on many points:

  * Eq 4:  I don't think there are any guarantees that the moved vector, $x + \epsilon(W_{c'} - W_c)$, is still in the local linear region of the CNN function around $x$.  So, while moving in this direction will result in a larger logit for $c'$ when extending the linear function like this, the original CNN function may or may not classify the new $x$ as $c'$.  How much does this matter?  The issue is somewhat addressed in the next section 4.4 (that that wasn't clear initially); but even there, fixing the number of steps could still force some points to go outside the local linear region of the previous sample.

  * Sec 4.4:  This describes a method for finding bits to flip in a binary image, but not how to apply the method to continuous image inputs (e.g., finding the point outside a ball centered on the current sample that minimizes the logit distance supposing local linearity.

  * This suggests a theoretical question I'm not sure of the answer to offhand:  Given x and y, does there have to exist a path x1,...,xn of finite length between the two points where each $x_{i+1}$ is in the local linear region of $x_i$?  And if not, under what conditions will that be the case?


In the overall picture of the paper, the interpolation method is never used to suggest an alternate measurable definition of "capacity" or "complexity" as described in the Sec 2 intro.  Can it be used like this?  That would tie together the paper much more.



Additional questions and technical points that I found unconvincing on the first part (though, as I mentioned above, I think these experiments don't add much to the overall set of contributions):

* If the architecture is always alternating (conv, pool, conv, pool, ...), then the depth can't be varied without also changing the input or feature map size.  So the depth is not varied independently of feature map spatial size or number of first fc layer connections.

* Figs 9, 10:  The large noisy curve upwards in loss but not error seems related to the choice not to use any regularization.  Adding a small amount of weight decay (L2 regularization) may prevent this, for example.  Since this is usually used in practice, I think it would make sense to use here as well.








Other smaller comments:


* p.2 para2:  piecewise linaer with relu was used in alexnet (Krizhevsky et al), while resnet (He et al) introduced residual connections to cnns.  which do you mean here?

* 2.2 While the idea of flatness is certainly important, I found the section may be too detailed on pac-bayes --- how is this related to the rest of the paper?

* 2.4 "interpolates a sequence of elements ... that least change the output"  -- sounds like a geodesic?  but output value and path of min curvature could be different things.  nevertheless, is there a view of the interpolation path along these lines?


* The norm on p.14, written in the transpose form, describes a convolution-deconvolution similar to the "deconv" net used in Zeiler & Fergus 2014; there may be a difference from that work, though, which is that $\Delta_1$ is a change vector and not a set of activations.


* does not address CNNs with residual connections --- note that generalization conflicting with LRT applies to both.  transformers are mentioned, though, with acknowledgement that they are not studied in this work, which I think is OK.


* Sec 4.2 "activities" -- "activations" is the more common term for this

* p.12 just below eq 3: broken link

---

> ### Author Response · Authors · 2025-03-13
>
> The reviewer requests a more focused presentation, and raises a number of issues to be addressed. We propose to do the following.
>
> - _It also lacks focus, essentially having two parts, the first experimental val measurements and the second the interpolation method. I don't think the first part adds much here, since these behaviors are already established including in some of the works already cited._
>
> We agree that the first part mostly confirms known results; we propose to move those to an appendix so as not to distract from the main innovation of our submission. We were not able to locate a reference clearly showing that MLPs do not benefit from additional hidden layers, but if a reviewer could point us to such literature we will replace the planned appendix with a reference.
>
> - _Eq 4: I don't think there are any guarantees that the moved vector, is still in the local linear region of the CNN function. … The issue is somewhat addressed in the next section 4.4 (that wasn't clear initially); but even there, fixing the number of steps could still force some points to go outside the local linear region of the previous sample._
>
> We agree that there is no guarantee, and this issue was the motivation for using average rather than max pooling. Informal experiments showed max pooling to enlarge the linear regions and thus improve our approximation. Since Reviewer hxmB has also raised this issue, we will repeat those experiments more formally and add an appendix reporting on the accuracy of our approximation.
>
> - _This suggests a theoretical question I'm not sure of the answer to offhand: Given x and y, does there have to exist a path x1,...,xn of finite length between the two points where each is in the local linear region of? And if not, under what conditions will that be the case?_
>
> This certainly is an interesting question, but we unfortunately do not know of an efficient way to answer it. We therefore propose to describe it as an avenue for future research in the conclusion.
>
> - _In the overall picture of the paper, the interpolation method is never used to suggest an alternate measurable definition of "capacity" or "complexity" as described in the Sec 2 intro. Can it be used like this? That would tie together the paper much more._
>
> Our intuition is that the notion of “inductive bias” is in fact more general than that of “capacity”. We propose to state this more explicitly in the conclusion.
>
> - _If the architecture is always alternating (conv, pool, conv, pool, ...), then the depth can't be varied without also changing the input or feature map size. So the depth is not varied independently of feature map spatial size or number of first fc layer connections._
>
> This limitation of our design will be pointed out.
>
> - _Figs 9, 10: The large noisy curve upwards in loss but not error seems related to the choice not to use any regularization. Adding a small amount of weight decay (L2 regularization) may prevent this, for example. Since this is usually used in practice, I think it would make sense to use here as well._
>
> We included only the simplest options in our algorithm, to avoid the need for hyperparameter optimization. We agree that some weight decay would have been a good choice here, but decided against it given that generalization was not adversely affected.
>
> - _p.2 para2: piecewise linear with relu was used in alexnet (Krizhevsky et al), while resnet (He et al) introduced residual connections to cnns. which do you mean here?_
>
> We will correct the reference - Krizhevsky et al should indeed be cited here.
>
> - _While the idea of flatness is certainly important, I found the section may be too detailed on pac-bayes --- how is this related to the rest of the paper?_
>
> The relationship is not strong, and that subsection will be omitted.
>
> - _"interpolates a sequence of elements ... that least change the output" -- sounds like a geodesic? but output value and path of min curvature could be different things. nevertheless, is there a view of the interpolation path along these lines?_
>
> This is a very interesting insight, and we will add some discussion on this perspective.
>
> - _The norm on p.14, written in the transpose form, describes a convolution-deconvolution similar to the "deconv" net used in Zeiler & Fergus 2014; there may be a difference from that work, though, which is that is a change vector and not a set of activations._
>
> Another interesting connection that we missed, and will add in our resubmission.
>
> - _Sec 4.2 "activities" -- "activations" is the more common term for this_
>
> We will correct this everywhere.
>
> - _p.12 just below eq 3: broken link_
>
> Will be corrected.

---

### Review · Reviewer_hxmB · 2025-03-07

**Summary Of Contributions:**

The paper's primary goal is to propose and measure a new notion of inductive bias for convolutional neural networks and how network depth affects this inductive bias. At a high level, the proposed inductive bias is demonstrated with the observation that deeper CNNs tend to demonstrate smoother and more semantically coherent interpolation between different samples.

The method involves first computing the linear representations of the model locally around the data point and then gradually moving data point towards another point while minimizing changes in logits. The results show that deeper models, in general, exhibit interpolations that are smoother and more coherent. This intuition is partially validated with a new quantity called bit change summation.

**Audience:**

No

**Broader Impact Concerns:**

I do not see any ethical concerns for this paper.

**Claims And Evidence:**

No

**Requested Changes:**

**Additional material**

- Revision and further explanation according to weakness above.
- Pseudocode for interpolation.
- Show evidence that the proposed method explains generalization better than existing methods like PAC-Bayes.
- Mathematical formula for bit change summations instead of plain text.

**Typo**

- Bottom of page 15: "shallower networks have more low-frequency components" should be high frequency?

**Strengths And Weaknesses:**

Overall, this submission presents some interesting needs but needs more evidence to support the claims being made.

**Strength**

- The proposed analysis and techniques are original and interesting.
- The methodology and intuition are well-explained.

**Weakness**

- The experimental results are on the simpler side. The experiments are done with fairly simple CNNs on binarized MNIST. In this setting, PAC-Bayes bounds are actually fairly tight. They only become looser in more realistic and complex settings. While there is nothing wrong with a simple setup for clear understanding, the claim that this framework could better explain the generalization of CNNs is, in my opinion, an overclaim. In order to substantiate the claim, the authors should convert the observation into a single scalar that can be used to compare models of different hyperparameters and follow protocols like [1,2] to verify its effectiveness.
- Some aspects of the papers are not very well-motivated. For example, sharp minima are given a large amount of exposition but it is not clear to me how sharp minima are related to the proposed idea. The same goes for the bias-variance trade-off. If there are necessary connections, I think they should be stated explicitly.
- Another idea that is not well-motivated is the idea of a converter between metric spaces. It is not clear to me why this terminology is necessary. It is also not clear to me why it is important that they are metric spaces other than the fact you can measure distance by virtual of being real vectors. Even in that case, it is not clear to me why it is desirable to measure distance in the logit space rather than something like KL divergence or total variation distance in the output probability.
- The proposed algorithm also lacks some details and explanation, which I believe would benefit from having a pseudocode and further analysis. For example, when picking $x_1$, is it guaranteed that $x_1$ would also be in the same linear region of input space? If not, wouldn't that violate the approximation?
- The results in figures 16 and 17 do not demonstrate that deeper models always have smaller pixel-wise sum (e.g., depth 3 and 4). It would be good to have a single summarizing number to tell any two models apart.
- The claims in the paper should be falsified. In the sense that does the framework make any prediction that you have not observed and does the prediction agree with reality? The simplest thing to do would be to have a different architecture (e.g., skip connection) or different optimizers.


**reference**

[1] Fantastic Generalization Measures and Where to Find Them. Jiang et al.

[2] In search of robust measures of generalization. Dziugaite et al.

---

> ### Author Response · Authors · 2025-03-13
>
> The reviewer raises several valid points, and suggests two references which will substantially improve our submission. We propose to address these points as follows.
> - _… the claim that this framework could better explain the generalization of CNNs is, in my opinion, an overclaim. In order to substantiate the claim, the authors should convert the observation into a single scalar that can be used to compare models of different hyperparameters and follow protocols like [1,2] to verify its effectiveness._
>
> Computational constraints unfortunately prevent us from applying our methods on more challenging problems. However, conversion to a single scalar should be straightforward (as a sum over classes and interpolants), and we will use that to implement a simple version of the protocols in [1,2] for our tasks, and some additional architectures (see below).
>
> - _Some aspects of the papers are not very well-motivated. For example, sharp minima are given a large amount of exposition but it is not clear to me how sharp minima are related to the proposed idea. The same goes for the bias-variance trade-off. If there are necessary connections, I think they should be stated explicitly._
>
> Our background on sharp minima was intended as a sample of related work; however, the reviewer's [2] is a much better representative of such work, and we propose to replace the discussion on sharpness with a brief summary of [1] and [2]. The bias-variance trade-off is discussed as a core idea in conventional learning theory, and we will contextualize that section more clearly in our introduction and conclusion.
>
> - _Another idea that is not well-motivated is the idea of a converter between metric spaces. It is not clear to me why this terminology is necessary. It is also not clear to me why it is important that they are metric spaces other than the fact you can measure distance by virtual of being real vectors. Even in that case, it is not clear to me why it is desirable to measure distance in the logit space rather than something like KL divergence or total variation distance in the output probability._
>
> This idea helped us formulate our approach, but in retrospect we agree that the eventual findings do not benefit significantly from that section. We propose to replace it with a brief motivation for working in logit space (while admitting that other distance measures could serve the same purpose).
>
> - _The proposed algorithm also lacks some details and explanation, which I believe would benefit from having a pseudocode and further analysis. For example, when picking, is it guaranteed that would also be in the same linear region of input space? If not, wouldn't that violate the approximation?_
>
> We will add pseudocode to better explain our algorithm. The need to (mostly) stay within a given linear region is indeed a subtle matter; please see our response to Reviewer PVKq below, including additional measurements we propose.
>
> - _The results in figures 16 and 17 do not demonstrate that deeper models always have smaller pixel-wise sum (e.g., depth 3 and 4). It would be good to have a single summarizing number to tell any two models apart._
>
> Agreed, as stated above we will report on such a scalar in the revised version of our submission.
>
> - _The claims in the paper should be falsified. In the sense that does the framework make any prediction that you have not observed and does the prediction agree with reality? The simplest thing to do would be to have a different architecture(e.g., skip connection) or different optimizers._
>
> We will implement an efficient approximation of our algorithm in order to carry out such contrastive experiments, reporting the scalar measure suggested in this review.
>
> - _Mathematical formula for bit change summations instead of plain text._
>
> Will be added.
>
> - _Bottom of page 15: "shallower networks have more low-frequency components" should be high frequency._
>
> Will be corrected.

---

### Decision · Action_Editor_7dP8 · 2025-04-21

**Recommendation:** Reject

**Comment:**

The paper is studying an interesting topic on CNN inductive bias, and proposes some potentially interesting interpolation method. However, the reviewers found this method to be underdeveloped. Overall, the quality of the writing, experimental design, and experimental scope needs to be substantially improved. The authors should also make sure that the claims made are supported with clear theoretical or empirical evidence.

Reviewers encouraged the authors to resubmit after incorporating feedback, specifically suggesting they refocus the work on the more interesting interpolation aspects, further substantiate arguments (e.g., on local linear region chaining), and design experiments more carefully, e.g. varying depth without also varying the spatial extent of the feature map or parameters of a downstream FC layer (for example, stacking convolutions only within one pooling block).

**Audience:**

The underlying topic (understanding inductive bias in CNNs, and how depth affects it) is of interest to some of TMLR audience. However, the paper itself needs to be significantly improved.

**Claims And Evidence:**

The reviewers agreed that the claims made in the submission are not supported by accurate, convincing and clear evidence. In particular, the experimental design was found to be overly simplistic. The experiments were carried out on very simple setups (binary MNIST), but the actual claims made (e.g., about MLP performance)  were quite broad and unsubstantiated given the scope of the experiments. The experiments also did not account for factors such as regularization or different architectures. Finally, some of the claims made (e.g., deeper models having smaller pixelwise sums) were not even consistently supported by the figures in the paper.

**Resubmission Of Major Revision:**

The authors may consider submitting a major revision at a later time.